# The Psychometric Function for Focusing Attention on Pitch

**Adam Reeves** 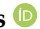

Department of Psychology, Northeastern University, Boston, MA 02115, USA; reeves@neu.edu

**Abstract:** What is the effect of focusing auditory attention on an upcoming signal tone? Weak signal tones, 40 ms in duration, were presented in 50 dB continuous white noise and were either uncued or cued 82 ms beforehand by a 12 dB SL cue tone of the same frequency and duration as the signal. Signal frequency was either constant for a block of trials or was randomly one of 11 frequencies from 632 to 3140 Hz. Slopes of psychometric functions for detection in single-interval (Yes/No) trials were obtained from three listeners by varying the signal level over a 1–9 dB range. Plots of log(d') against signal dB were fit by linear functions. Slopes were similar whether signal frequency was constant or varied, as found by D. Green. Slopes for uncued tones increased by 14% to 20% more than predicted by signal energy (i.e., 0.10), as also found previously, whereas slopes for cued tones followed signal energy corrected for an 8 dB sensory threshold. That pre-cues help attention focus rapidly on signal frequency and permit listeners to act as near-ideal detectors of signal energy, which they do not do otherwise, supports a key hypothesis of Grossberg's ART model that attention guided by conscious awareness can optimize perception.

**Keywords:** attention; audition; energy; attention

## 1. Introduction

How well do listeners focus attention on a particular signal frequency? Green [1] and others [2–6] have shown that the threshold for detecting pure tones in broadband noise is higher, by about 3 dB, when listeners do not know at what frequency the signal will be presented than when they do know. The current research was undertaken to elucidate this improvement in perception due to foreknowledge. Here, the term 'frequency' describes the pitch in Hz, and 'level' describes the amplitude in decibels (dB). The term 'focusing attention' describes both the listener's objective task and his or her phenomenal awareness of the task, but not, in general, what focusing does. A general scheme for how attention can prime a signal by suppressing unwanted information is provided by Grossberg's ART theory [7] (p. 18); see Dresp-Langley [8]. ART requires the signal to be learnt, as resonance and top-down matching with memory is required, and in the psycho-acoustic literature cited here, this is the case: only experienced listeners who have memorized the possible tones are used. In the present context, it is possible to formulate a specific hypothesis, namely, that knowing the signal frequency allows the listener to focus attention in advance on the signal's *critical band* (CB), the band of tones around the signal which interact with it [9]. Focusing on the signal CB suppresses noise from non-signal CBs and so increases detectability (d') as compared to not knowing the signal in advance [10], a form of suppression which has been investigated physiologically in primates [11].

Experimentally, providing the same signal in every trial permits the listener to focus attention on the signal CB, whereas varying the signal at random across trials does not. Green [1] compared these two conditions, which I will term *const* when signal frequency is constant for a block of trials and *var* when signal frequency is varied. (In *var*, signal frequency is typically selected at random across trials from between 5 and 22 possible frequencies, each at least one CB from the next.) The listener's task in both *const* and *var* is to detect weak, brief (<350 ms) signals in continuously present wide-band noise that covers the range of possible signal frequencies. Such wide-band noise is convenient

in that it elevates individual listeners' thresholds to the same level, within a dB or so, over a wide range of frequencies, making the experimental measurements possible in uniform conditions; this is not so for thresholds in silence, which vary idiosyncratically over individuals and frequencies.

How well attention can be focused in *const* has also been determined using the 'probe-signal' method of Greenberg and Larkin [12]. In this method, which differs from Green's, so-called 'probe' tones are occasionally presented at unexpected frequencies above or below the (constant) signal frequency. Signals and probes are near the threshold, and, apart from their frequencies, they are identical in duration and quality. Probes inside the signal CB are heard in proportion to their distance from the signal frequency, defining an 'attention band' around the signal [13–17]. Distant probe tones, those outside the signal CB, are not heard, being attenuated by up to 8 dB [17]. Although the close match between the attention band and the CB fails below 500 Hz, when the attention band more closely follows a narrower auditory filter [14], the argument of this paper will be phrased in terms of CBs as signals below 632 Hz were not used. When tones are very brief (20 ms), focusing fails to exclude neighboring CBs, and the attention band widens and peaks just below the signal frequency [15], but with longer durations, focusing on single CBs is successfully accomplished.

Proof that focusing primarily suppresses distant probe tones rather than enhancing the signal tone was provided by Scharf, Magnan, and Chays [18], who compared thresholds before and after vestibular neurotomy. This operation randomly severs the olivio-cochlear bundle, which mediates cortical feedback to the outer hair cells of the cochlear [19,20]. After neurotomy, patients lost the ability to suppress probes away from the signal CB but showed no change in signal threshold [18]. (Interestingly, the hearing of speech in noise is unaffected, speech being broadband so there is no particular 'signal' frequency.) Here, I assume that the suppression of non-signal CBs is the primary effect of attentional focusing, although Tan et al. [21] also reported a minor 2 dB signal enhancement due to focusing.

When a wide-band noise is applied, the listener who can focus on the signal CB (in *const*) will suppress noise from non-signal CBs, as evidenced by the probe-signal data just discussed, but a listener who attends to all possible signal frequencies (in *var*) cannot suppress noise, as all CBs potentially contain a signal. Thus, the detection mechanism in *var* will sum more noise than that in *const*, and the detectability of the signal (d') in *var* will fall below that in *const*.

Note that noise suppression may be total, exemplifying 'exclusion' in the terms of Lu and Dosher [22], or partial, exemplifying 'attenuation', as in Treisman [23]. As Green [1] pointed out, given the wide range of frequencies he employed, excluding all the noise in *var* predicts a 10 dB loss relative to *const*, not the 3 dB loss he obtained. Green's suggested explanation for this discrepancy was that listeners fail to focus completely on the signal CB even in *const*, so noise from non-signal CBs is attenuated rather than excluded.

One aim of the current research was to test Green's suggestion by validly pre-cuing the signal frequency. Frequency *var* versus *const* was crossed with validly cueing versus not cuing, a *var/const × valid cue/no cue* design adopted from Richards and Neff [24]. Validly cuing the signal frequency helps the listener focus on the signal frequency [2,10,16,25], so any uncertainty about the signal frequency should be reduced, perhaps even eliminated, by cuing.

Note: Richards and Neff [24] had crossed frequency certainty with cuing, as in the current study. They used an 'informational mask' consisting of a multitone array of tones all outside the signal CB. In *const*, the signal was always 1000 Hz, and the mean benefit of a valid pre-cue was 6.5 dB compared to no cue. In *var*, the signal was a random one of five tones, and the benefit of a valid pre-cue of the same frequency as the upcoming signal averaged 13.5 dB. They argued that attention can be focused within 50 ms, as longer cue-signal ISIs hardly increased the effectiveness of the cue. They did not measure slopes, but their cue effect (in dB) was encouragingly large. However, a multitone masker encourages attention to focus on non-signal CBs, as shown by their additional finding that pre-cuing

the mask array helped the listener re-direct attention away from the mask and greatly aided detection. This would not apply to the broadband noise used here and Green [1].

Cues tones presented very close to the signal are not only informative but also interfere with detection [26,27] even when the cue is valid (i.e., has the same frequency as the signal). In the present work, valid cues were presented 82 ms before the signal, when interference is small, about 2 dB in both *const* and *var* [28].

*Previous Studies: Energy Detection*

The experimental literature contains several previous studies of the role of attention in detection, starting with Green [1], and I analyze these in the next section. (I provide new results from three listeners in the *var/const* × *valid cue/no cue* experiment.) My analyses show that the widely assumed 'energy detection' model of auditory threshold provides a rather poor approximation of the data.

Green [1] measured proportion correct detection (Pc) in two-alternative forced choice (2AFC) trials, for 800, 1250, 2250, and 3200 Hz signal tones, which were constant in each block of trials (*const*), and for 100, 300, 500, 1000, or 3500 Hz signal tones, randomized across trials (*var*). Cues were never provided. Tones were 100 ms in duration (and ramped on and off to avoid clicks, as is standard). Detection was measured over an 11 dB range of signal levels. Tones were presented in wide-band noise whose amplitude was constant at a 40 dB spectrum level. Pc's from all frequencies taken together are plotted in Figure 2 of his article.

Sound level in decibel (dB) units equals $10\log_{10}[(P/Po)^2]$, where Po is a reference level, either 0.0002 dynes/cm$^2$ in the case of dB SPL (sound pressure level) or the threshold in the case of dB SL (sensation level). Since thresholds in dB SPL varied with frequency, Green [1] plotted 2AFC accuracy (Pc) against signal dB SL, where at every frequency, 5 dB SL was defined to correspond to Pc = 75% (chance being 50%). Given the small numbers of recorded observations at each signal level, from 3 to 6, median (rather than mean) Pc's were read from Figure 2 in his work and are given below in Table 1 under the heads Pc *const* and Pc *var*.

**Table 1.** Data from Green [1], Figure 2. Pc const and Pc var are medians at each signal dB level. Amp = $10^{dB/20}$ is given for reference. Noise level was constant. Pcon and Pvar (converted to d'con and d'var) are the Pc's corrected for 2% lapsing.

| Amp | dB | Pccon | Pcvar | Pcon | Pvar | d'con | d'var | log d'con | log d'var |
|------|-----|-------|-------|-------|-------|--------|--------|-----------|-----------|
| 1.12 | 1 | 0.54 | 0.55 | 0.541 | 0.551 | 0.145 | 0.181 | −0.839 | −0.741 |
| 1.26 | 2 | 0.55 | 0.58 | 0.551 | 0.582 | 0.181 | 0.291 | −0.741 | −0.535 |
| 1.41 | 3 | 0.64 | 0.66 | 0.643 | 0.663 | 0.518 | 0.596 | −0.286 | −0.225 |
| 1.58 | 4 | 0.66 | 0.68 | 0.663 | 0.684 | 0.596 | 0.676 | −0.225 | −0.170 |
| 1.78 | 5 | 0.78 | 0.76 | 0.786 | 0.765 | 1.120 | 1.023 | 0.049 | 0.010 |
| 2.00 | 6 | 0.87 | 0.82 | 0.878 | 0.827 | 1.644 | 1.330 | 0.216 | 0.124 |
| 2.24 | 7 | 0.91 | 0.91 | 0.918 | 0.918 | 1.972 | 1.972 | 0.295 | 0.295 |
| 2.51 | 8 | 0.97 | 0.93 | 0.980 | 0.939 | 2.893 | 2.184 | 0.461 | 0.339 |
| 2.82 | 9 | 0.98 | 0.97 | 0.990 | 0.980 | 3.279 | 2.893 | 0.516 | 0.461 |
| 3.16 | 10 | 0.98 | 0.95 | | | | | | |
| 3.55 | 11 | 0.98 | 0.98 | | | linear | slope | 0.177 | 0.146 |

For signals above 9 dB SL, the mean error rate (E) did not depend on level, so it is likely that the few remaining errors, which averaged 2%, were due to lapses in attention [29,30]. Corrected for lapsing with E = 2%, the detection rates (Pc − 0.5E)/(1 − E) are listed in Table 1 under the headings Pconst and Pvar. Plotting these against signal dB SL yields slopes of 0.056 in *var* (r = 0.98) and 0.062 in *const* (r = 0.99), close to the slopes of 5% per dB reported in the auditory detection literature [5,6,31]. Table 1 gives d'const = $\sqrt{2}z(\text{Pconst})$ and d'var = $\sqrt{2}z(\text{Pvar})$, where z(P) is the standard Normal z-score of proportion P. The final two columns give these d's in $\log_{10}$ units. (Note: log(d') exists as all Ps exceeded 50%, or d' = 0.)

According to Green and Swets [32], a detector of sinusoidal signal tones obeys

$$d' = k(S/No) \tag{1}$$

where S is the signal level, No is the noise level in the channel that detects the signal, and k is a constant of proportionality. They took No to equal the external noise provided by the experimenter, because there was no good evidence for internal auditory noise, and Brownian motion in air ensures No > 0 and so prevents division by zero in the quiet. For a *peak–trough* detector, S and No are in units of amplitude or sound pressure, P/Po. For an *energy* detector, S and No are in units of $(P/Po)^2$. Converting Equation (1) to dBs by canceling Po in the ratio S/No, and writing $S_{dB}$ and $No_{dB}$ for the Signal and Noise in dBs,

$$\log_{10}(d') = \log_{10}(k) + b(S_{dB} - No_{dB}) \tag{2}$$

where the slope, b, is 1/20 for a peak–trough or amplitude detector and 1/10 for an energy detector [32].

The regression of $\log_{10}(d')$ on $S_{dB}$ from Table 1 gave slopes (b) of 0.146 in *var* (r = 0.98) and 0.177 in *const* (r = 0.98). These slopes clearly reject the peak–trough detector (b = 1/20) but are also on average 16% steeper than the energy detector. Green [1] stated that the slope was 1/10 in both *var* and *const*, but he may have been misled, both because log(d′) puts undue weight on near-zero d′s, and because d′ becomes unstable at high Pc's [30]. However, Green's claim has entered the literature and it is often taken for granted that his data showed that the ear is an energy detector.

Green, Birdsall and Tanner [33] had previously used uncued 1000 Hz signals in *const* and again reported slopes consistent with energy detection in wide-band noise for four listeners and three stimulus durations. However, interpolating the $S_{dB}$ levels at d′ = 0.5 and d′ = 2.0 in all 12 of their plots, the average of the resulting log(d′) versus $S_{dB}$ slopes is again 14% steeper than the energy detector. Dai [31] also reported slopes of 0.14 in *const* and 0.15 in *var*, not 0.10, using an uncued 'profile' task in which listeners discriminated an array of 21 well-spaced tones from the same array plus a signal tone. It thus appears that for uncued tones of constant frequency, the log-log psychometric slopes are consistently around 14% too steep for pure energy detection.

As stated above, Green [1] concluded from his data that the listeners are uncertain about signal frequency, not only in *var* but also in *const*, because the thresholds in *var* and *const* were only 3 dB apart, not the 10 dB estimated from noise exclusion. The weakness of the uncertainty effect is too dramatic to be an artefact of the somewhat different frequency ranges Green employed in *const* and *var*. However, to obtain a steepening of 14% in *const* requires the listener to be uncertain about which of at least 32 channels contains the signal [29,32]. Such high channel uncertainty seems unlikely, given that the signal in *const* was fixed in every auditory parameter (lateralization, duration, onset time, and frequency). Alternatively, the noise level in *var* might be determined by the maximal noise in each CB, not the summed noise across CBs, an idea which correctly predicts a 3 to 4 dB uncertainty effect [32]. However, attention to all the possible signals in *var* implies attending to the noise in each of the possible signal CBs, rendering the max operator unrealistic. Scharf, Reeves, and Giovanetti [34] offered yet another explanation of the weak uncertainty effect. Attention can be focused on an unexpected signal frequency in less than 52 ms [24,28], and as Green [1] and Dai [31] used 100 ms tones, much of the noise from non-signal CBs in *var* could be excluded by shifting attention to the signal before it terminated. That is, rather than assuming uncertainty in *const*, Scharf et al. [34] assumed more certainty in *var*. They [34] estimated the true uncertainty effect as 9 dB in an overshoot experiment in which attentional focusing was completely disrupted by the onset of broadband noise, close to the predicted 10 dB. Thus, the conclusion that the normal listener excludes noise from all non-signal CBs when focusing may be correct after all.

The uncertainty effect in the current experiment was expected to be 3 dB, as the *const/var* method was used, rather than the overshoot procedure. It is the slopes that are

of concern here. Given the earlier results [1,31,33], it seemed likely that with no cue, the present results would also show a steeper psychometric slope than the ideal energy detector. The question at issue was whether pre-cuing could help listeners focus attention and bring the slope closer to 1/10, the energy detector. If so, the claim can be made that attention to known sounds permits the ear to operate as an ideal receiver of signal energy.

## 2. Methods

The terms *var* and *const* will continue to be used for conditions in which frequency was varied unpredictably over trials or was constant for a block of trials. The *var* and *const* conditions were like Green's [1], except that tone duration was 40 ms, not 100 ms. The same frequencies were used in both conditions, since the uncertainty effect decreases with frequency [25]. Signals were preceded 82 ms earlier by a valid cue, also 40 ms in duration, or were, like Green's, uncued.

*Participants.* One male (MA) and two female (TA and NA) Northeastern University undergraduates, aged 19, 20, and 22, served as listeners. All three had normal audiograms and detection thresholds. Hour-long sessions were run over several weeks to obtain data. None reported using drugs (prescribed or otherwise) during the course of the study. The study was authorized by the human subjects committee of Northeastern University. Listeners gave informed consent. They were paid USD 10 per hour and were told they could leave the study at any time without loss of payment. They were informed that the study was undertaken to facilitate audiometry, but not that it was a study of focusing.

*Apparatus.* Listeners sat in a sound-attenuated booth (Eckel Industries) and heard sounds generated by a Tucker-Davis (Alachua, FL, USA) TDT System III signal processor (RP2.1) sampled at a rate of 48.83 kHz. A microcomputer (Dell Optiplex GX270; Dell Computers, Round Rock, TX, USA) programmed in Pascal controlled the processor and collected data via a response box (TDT BBOX). Sounds were sent through a headphone driver (TDT HB7) to Sony MDR-V6 cushioned headphones (Sony Corp, Tokyo, Japan). Waveforms, frequency content, and distortion were checked with a wave-analyzer and an oscilloscope. Digital filters were used to generate new wide-band 50 dB SPL noise on every trial, which resembled an analogue bi-quad bandpass filter flat from 200 to 6000 Hz.

*Stimuli.* Trials began with a warning signal appearing on a visual display screen. Half a second later, a 40 ms cue tone appeared in 'cued' trial blocks. In 'no-cue' blocks, the cue was set to zero amplitude to maintain timing by the program. The cue or silent cue interval was followed after 82 ms by a sinusoidal tone of 40 ms duration in half the trials, or no signal in the remaining trials. The 40 ms cues and signals were gated by cosine ramps (5.6 ms rise and 6.4 ms fall ms times), so each was 52 ms *in toto*.

Tone duration was 40 ms to reduce the chance that, in *var*, the listener could shift attention during the signal. An even briefer tone might reduce this chance even further, but at a cost; the attention band matches the critical band (the CB) for long duration tones but is considerably wider for 5 ms tones [17] and still somewhat wider even for the 40 ms tones used here [15]. Thus, to keep the attention band within reasonable limits, signal duration was not reduced to below 40 ms. The spectrum level of the 50 dB SPL broadband noise, namely 12.44 dB from 570 to 3400 Hz, was also chosen to be low since the attention band, unlike the CB, widens at higher levels [13].

*Procedure.* Signal and no-signal trials were intermixed at random, and listeners reported whether the signal was present or not (a single interval 'Yes/No' task). Blocks comprised 110 trials each. The cue condition was alternated after every trial block. A single-interval method was used rather than 2IFC to ensure that the cue signal interval was the same on every cued trial. Unless voluntarily delayed by the listener, the next trial began 500 ms after the response. In *const*, the same frequency tone was presented in every one of 55 signal trials. In *var*, the signal tone was selected at random from the same list of 11 frequencies as were employed in *const*, with each frequency appearing 5 times. These frequencies were spaced at least one CB apart [9].

*Thresholds*. To accommodate slight variations in sensitivity across sessions, the level of the middle (1266 Hz) tone was adjusted in 1 dB increments to reach 89% correct at the start of each session. All other signal tones were adjusted by adding the same amount to each listener's no-cue thresholds. These were measured in 3 initial sessions, for the 50 dB noise, at each of the 11 frequencies from 632 to 3140 Hz, using an adaptive procedure that converged on 79% correct. Table 2 lists the 11 frequencies and mean thresholds in dB SPL for each listener. (The expected increase in threshold from 632 to 3140 Hz, from the ratio of their ERBs, is 5.9 dB; actual increases for NA, TA and MA were 4.1, 7.3, and 3.3 dB.) The mean threshold at 1082 Hz was 35 dB, close to the 39 dB found by Baer, Moore and Glasburg [35] for 40 ms, 1000 Hz tones heard in background noise that was 3 dB higher than that used here. When presented, cues were 12 dB above the levels in Table 2. Experimental blocks were run after the levels were adjusted. At the start of each block, five additional (unrecorded) trials were run to notify the listener of the current condition.

**Table 2.** Thresholds of 40 ms tones in 50 dB SPL noise for each listener and frequency.

| Hz | 632 | 767 | 917 | 1082 | 1266 | 1481 | 1720 | 1994 | 2318 | 2693 | 3140 |
|----|-----|-----|-----|------|------|------|------|------|------|------|------|
| NA | 36.2 | 36.3 | 36.3 | 36.3 | 36.1 | 36.0 | 36.8 | 37.6 | 38.4 | 39.2 | 40.3 |
| TA | 31.6 | 32.4 | 33.5 | 33.2 | 33.4 | 35.5 | 36.7 | 36.1 | 37.5 | 37.3 | 38.9 |
| MA | 36.2 | 36.6 | 36.9 | 37.2 | 37.6 | 38.0 | 38.3 | 38.6 | 38.9 | 39.2 | 39.5 |

*Design.* The four conditions obtained by crossing const versus var with no cue versus pre-cue were run on each of the 11 frequencies, at five signal levels ($S_{dB}$) spaced 2 dB apart. The order of signal levels, and of frequencies in *var*, was randomized within blocks. The order of conditions was randomized across blocks. Each listener ran in hour-long sessions over several weeks for a total of 60 blocks or 6600 experimental trials. The first week was devoted to obtaining no-cue signal thresholds. There followed two weeks of practice, during which the listeners became familiar with the Yes/No task, and with both cue and no-cue conditions in both *const* and *var*, before the experiment was run.

## 3. Results

Hit and false alarm rates in each trial block were converted to d' = z(Phit) − z(Pfa). In *const*, hit and false alarm rates were recorded for each frequency in each trial block. In *var*, the common false alarm rate was applied to each frequency; only the hit rates were frequency-specific. Detectabilities based on this assumption correlated well (r = 0.98) in pilot work with 2AFC detectabilities over the range of frequencies used, when measured with no cue, implying that listeners adopted the same criteria independent of frequency. This is not surprising since in *var*, the upcoming frequency was unknown. Note that 2AFC trials are normally preferred but cannot be used with a cue without unbalancing the two intervals.

The values of d' were tabulated for each signal level for each listener, signal frequency, and condition. Individual d's were between 0.03 and 4.0 (as logs, between −1.52 and +0.60), so errant floor and ceiling effects are possible. The d's and signal levels were therefore averaged into frequency groups, low (632–917 Hz), middle (1082–1720 Hz), and high (1994–3140 Hz), no consistent variation with frequency being apparent within each group. The rows in Table 3 give $\log_{10}(d')$, dB SPL, and dB SL, for each listener and frequency group (low fr, mid fr, and hi fr). The condition is specified by column headings from left to right, Cue *var*, Cue *const*, No cue *var*, and No cue *const*. Signal level in dB SL was obtained by subtracting from dB SPL the shifts given in Col. 1 below each listener. Thus, for TA low fr. in data row 1, dB SPL in cue var (row 1, col 2) was 38.90 and the shift (row 2, col 1) was 34.08, so dB SL (row 1, col 4) was 38.90 − 34.08 = 4.82. For TA low fr. cue const, dB SPL (row 1, col 5) was 39.0 and the shift (row 3, col 1) was 32.11, so dB SL (row 1, col 7) was 39.0 − 32.11 = 6.89, and so forth.

**Table 3.** Column 1 identifies the listener and frequency group. Successive columns identify dB SPL, log(d′), and signal dB SL, under the headings for the condition (cue var, cue const, no cue var, and no cue const). d′s were obtained at 5 or 6 signal levels with the cue, and 4 or 5 signal levels with no cue.

| Listener | Cue var | | | Cue const | | | No Cue var | | | No Cue const | | |
|---|---|---|---|---|---|---|---|---|---|---|---|---|
| Fr.group | dB SPL | log(d) | dB SL | dB SPL | log(d) | dB SL | dB SPL | log(d) | dB SL | dB SPL | log(d′) | dB SL |
| TA Low | 38.90 | 0.34 | 4.82 | 39.0 | 0.51 | 6.89 | 38.00 | 0.41 | 3.90 | 35.00 | 0.58 | 4.21 |
| 34.08 | 36.90 | 0.28 | 2.82 | 37.0 | 0.37 | 4.89 | 34.90 | −0.08 | 0.80 | 33.00 | 0.55 | 2.21 |
| 32.11 | 34.90 | 0.18 | 0.82 | 35.0 | 0.27 | 2.89 | 32.90 | −0.01 | −1.20 | 31.00 | 0.29 | 0.21 |
| 34.10 | 32.90 | −0.21 | −1.18 | 33.0 | 0.08 | 0.89 | 29.90 | −0.72 | −3.20 | 29.00 | −0.54 | −1.79 |
| 30.79 | 30.90 | −0.25 | −3.18 | 31.0 | −0.12 | −1.11 | | | | | | |
| TA Med | 43.80 | 0.43 | 6.80 | 42.0 | 0.54 | 5.47 | 39.80 | 0.54 | 7.31 | 39.00 | 0.53 | 4.72 |
| 36.99 | 41.80 | 0.41 | 4.80 | 40.0 | 0.36 | 3.47 | 37.80 | 0.32 | 5.31 | 37.00 | 0.48 | 2.72 |
| 36.53 | 39.80 | 0.32 | 2.80 | 39.0 | 0.32 | 2.47 | 35.80 | 0.26 | 3.31 | 35.00 | 0.43 | 0.72 |
| 32.49 | 37.80 | 0.12 | 0.80 | 38.0 | 0.22 | 1.47 | 33.80 | −0.52 | 1.31 | 33.00 | −0.09 | −1.28 |
| 34.28 | 35.80 | 0.05 | −1.20 | 37.0 | −0.04 | 0.47 | | | | 31.00 | −0.77 | −3.28 |
| | 33.80 | −0.45 | −3.20 | | | | | | | | | |
| TA High | 47.96 | 0.46 | 8.43 | 45.0 | 0.47 | 3.59 | 43.96 | 0.58 | 9.51 | 45.00 | 0.58 | 4.73 |
| 39.53 | 45.96 | 0.41 | 6.43 | 43.0 | 0.26 | 1.59 | 41.96 | 0.44 | 7.51 | 43.00 | 0.54 | 2.73 |
| 41.41 | 43.96 | 0.38 | 4.43 | 41.0 | 0.07 | −0.41 | 39.96 | 0.34 | 5.51 | 41.00 | 0.38 | 0.73 |
| 34.45 | 41.96 | 0.17 | 2.43 | 39.0 | −0.45 | −2.41 | 37.96 | 0.19 | 3.51 | 39.00 | −0.40 | −1.27 |
| 40.27 | 39.96 | 0.06 | 0.43 | 37.0 | −1.02 | −4.41 | | | | 37.00 | −0.52 | −3.27 |
| | 37.96 | −0.18 | −1.57 | | | | | | | | | |
| NA Low | 42.28 | 0.41 | 4.56 | 44.0 | 0.55 | 6.02 | 38.28 | 0.54 | 2.01 | 37.00 | 0.38 | 2.39 |
| 37.72 | 40.28 | 0.31 | 2.56 | 42.0 | 0.50 | 4.02 | 36.28 | 0.24 | 0.01 | 35.00 | 0.23 | 0.39 |
| 37.98 | 38.28 | 0.11 | 0.56 | 40.0 | 0.24 | 2.02 | 34.28 | −0.77 | −1.99 | 33.00 | −0.33 | −1.61 |
| 36.27 | 36.28 | −0.44 | −1.44 | 38.0 | 0.05 | 0.02 | 32.28 | −1.50 | −3.99 | 31.00 | −0.77 | −3.61 |
| 34.61 | 34.28 | −0.12 | −3.44 | 36.0 | −0.28 | −1.98 | | | | | | |
| | 32.28 | −0.50 | −5.44 | | | | | | | | | |
| NA Med | 42.29 | 0.38 | 3.42 | 42.0 | 0.32 | 5.53 | 38.29 | 0.39 | 1.78 | 39.00 | 0.41 | 3.54 |
| 38.86 | 40.29 | 0.18 | 1.42 | 41.0 | 0.20 | 4.53 | 36.29 | 0.20 | −0.22 | 37.00 | 0.33 | 1.54 |
| 36.47 | 38.29 | 0.03 | −0.58 | 39.0 | 0.20 | 2.53 | 34.29 | −0.78 | −2.22 | 35.00 | −0.19 | −0.46 |
| 36.50 | 36.29 | −0.24 | −2.58 | 37.0 | 0.05 | 0.53 | 32.29 | −1.22 | −4.22 | 33.00 | −0.27 | −2.46 |
| 35.46 | 34.29 | −0.88 | −4.58 | 35.0 | −0.12 | −1.47 | | | | | | |
| | 32.29 | −0.58 | −6.58 | | | | | | | | | |
| NA High | 44.84 | 0.42 | 4.65 | 44.0 | 0.34 | 3.14 | 40.84 | 0.44 | 1.54 | 39.00 | 0.33 | 1.30 |
| 40.19 | 42.84 | 0.31 | 2.65 | 42.0 | 0.07 | 1.14 | 38.84 | 0.04 | −0.46 | 37.00 | −0.17 | −0.70 |
| 40.86 | 40.84 | 0.09 | 0.65 | 40.0 | 0.00 | −0.86 | 36.84 | −0.97 | −2.46 | 35.00 | −0.70 | −2.70 |
| 39.31 | 38.84 | −0.21 | −1.35 | 38.0 | −0.41 | −2.86 | 34.84 | −1.45 | −4.46 | 33.00 | −1.40 | −4.70 |
| 37.70 | 36.84 | −0.28 | −3.35 | 36.0 | −0.45 | −4.86 | | | | | | |
| MA Low | 38.57 | 0.44 | 3.47 | 41.0 | 0.44 | 8.23 | 40.57 | 0.55 | 7.06 | 39.00 | 0.55 | 7.00 |
| 35.09 | 36.57 | 0.41 | 1.47 | 39.0 | 0.39 | 6.23 | 38.57 | 0.38 | 5.06 | 37.00 | 0.52 | 5.00 |
| 32.77 | 34.57 | 0.15 | −0.53 | 37.0 | 0.32 | 4.23 | 36.57 | 0.38 | 3.06 | 35.00 | 0.47 | 3.00 |
| 33.50 | 32.57 | −0.59 | −2.53 | 35.0 | 0.22 | 2.23 | 34.57 | 0.01 | 1.06 | 33.00 | 0.32 | 1.00 |
| 32.00 | 30.57 | −0.90 | −4.53 | 33.0 | −0.08 | 0.23 | | | | | | |
| MA Med | 39.79 | 0.48 | 4.82 | 41.0 | 0.51 | 5.08 | 41.79 | 0.53 | 4.84 | 39.00 | 0.58 | 3.55 |
| 34.97 | 37.79 | 0.39 | 2.82 | 39.0 | 0.44 | 3.08 | 39.79 | 0.44 | 2.84 | 37.00 | 0.46 | 1.55 |
| 35.92 | 35.79 | 0.15 | 0.82 | 37.0 | 0.24 | 1.08 | 37.79 | 0.36 | 0.84 | 35.00 | 0.22 | −0.45 |
| 36.95 | 33.79 | −0.26 | −1.18 | 35.0 | −0.02 | −0.92 | 35.79 | −0.34 | −1.16 | 33.00 | −0.78 | −2.45 |
| 35.45 | 31.79 | −0.30 | −3.18 | 33.0 | −0.50 | −2.92 | | | | | | |

**Table 3.** *Cont.*

| Listener | Cue var | | | Cue const | | | No Cue var | | | No Cue const | | |
|---|---|---|---|---|---|---|---|---|---|---|---|---|
| Fr.group | dB SPL | log(d) | dB SL | dB SPL | log(d) | dB SL | dB SPL | log(d) | dB SL | dB SPL | log(d') | dB SL |
| MA High | 41.03 | 0.42 | 3.83 | 41.0 | 0.38 | 3.05 | 43.03 | 0.47 | 5.93 | 43.00 | 0.60 | 5.88 |
| 37.20 | 39.03 | 0.20 | 1.83 | 39.0 | 0.23 | 1.05 | 41.03 | 0.37 | 3.93 | 41.00 | 0.51 | 3.88 |
| 37.95 | 37.03 | −0.03 | −0.17 | 37.0 | −0.11 | −0.95 | 39.03 | 0.14 | 1.93 | 39.00 | 0.25 | 1.88 |
| 37.10 | 35.03 | −0.30 | −2.17 | 35.0 | −0.54 | −2.95 | 37.03 | −0.41 | −0.07 | 37.00 | 0.07 | −0.12 |
| 37.12 | 33.03 | −0.39 | −4.17 | 33.0 | −0.65 | −4.95 | | | | 35.00 | −0.34 | −2.12 |

Regressions of $\log_{10}(d')$ against $S_{dB}$, as in Equation (2), were conducted separately for each frequency group, to determine if the slopes varied systematically with frequency. They did not, in agreement with Green [1], as shown by the averaged slopes plotted in Figure 1. Critically, the mean slope without a cue was 0.16 in *const* and 0.19 in *var*, close to the 0.16 slope found in Green [1] and the 0.14 slope in Dai [31], whereas the slopes with a cue averaged 0.107 in *const* and 0.102 in *var*, both close to 0.10. These data confirm Green's suggestion that the listener is uncertain about frequency in *const*, when—as in his experiment—there is no cue to guide attention. The new result is that *with* a cue, the listener is very close to an ideal detector of signal energy (slope: 0.10).

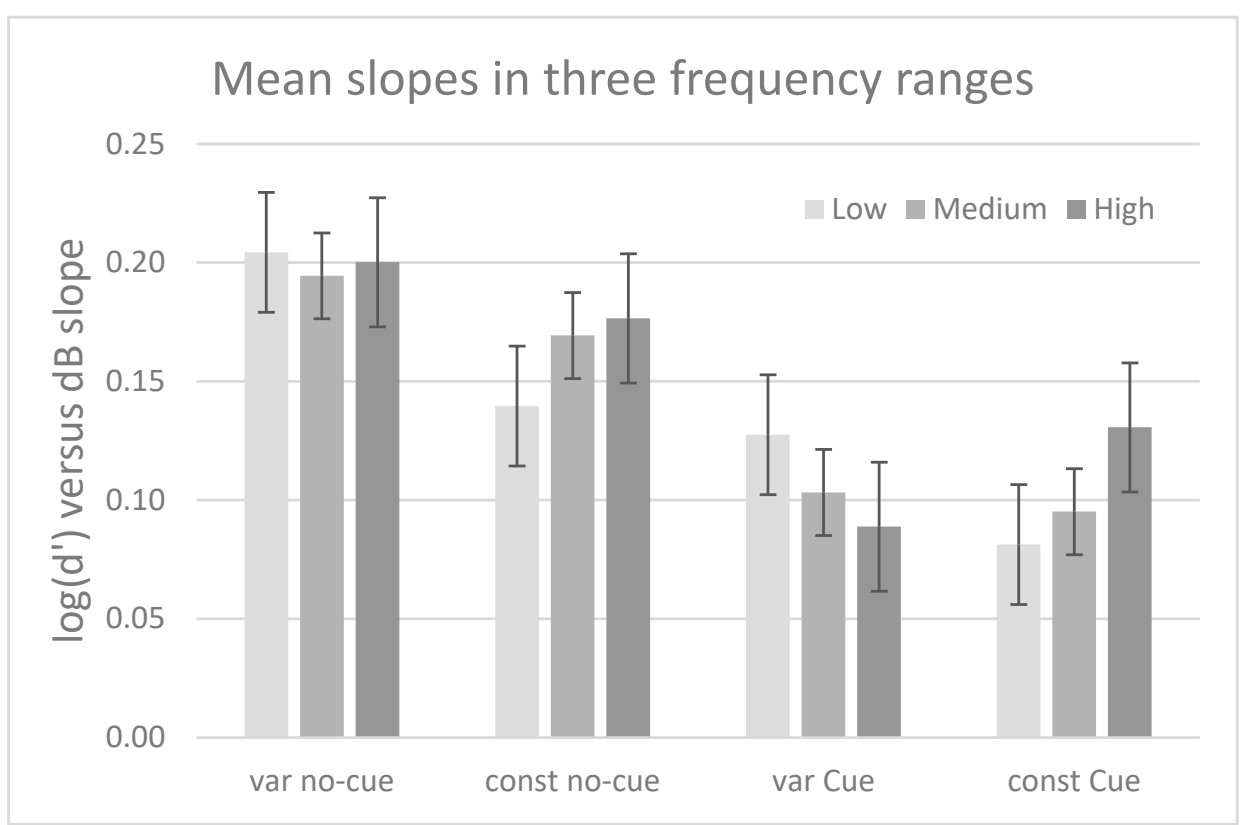

**Figure 1.** Mean slopes of log(d') versus dB in each condition: frequency uncertain ('var') or certain ('const') with no cue or validly cued 82 ms prior to the signal. Frequencies were in the low, middle, or high ranges (see text). Bars show ± SE of the mean.

This paper could stop here, given that the data—with a cue—conformed to Equation (1). However, data were also pooled across frequencies using the method of Green [1] with data shifted horizontally so d' = 1 ($\log_{10}$ d' = 0) at 0 dB SL. (Without shifting, the dependence of dB SPL on threshold scatters the data.) Shifts (given in Table 3, col 1) were obtained by dividing the intercepts by the slopes of the linear fits to the $\log_{10}(d')$ versus dB SPL data, separately for each listener, frequency group, and condition. Plots of Green's type

are shown in Figures 2–4 for listeners TA (top), NA (middle) and MA (bottom). Left panels show $\log_{10}$ (d') against dB SL in *var*—right panels, in *const*. There was no cue (upper panels for each listener) or a cue (lower panels). Solid lines show linear regressions following Equation (2). Data were fit with quadratic regressions (dotted lines), for which the proportionality predicted by Equation (1) is not quite correct, despite the high linear $r^2$ (see Table 4, col 5). The $r^2$ s in the last two columns of Table 4 are lowered by the additional variability from shifting, but still show that the quadratic $r^2$ exceeds the linear $r^2$ by up to 12%. It is unlikely that the quadratic fits were random as all the bows faced downwards and had the same general form.

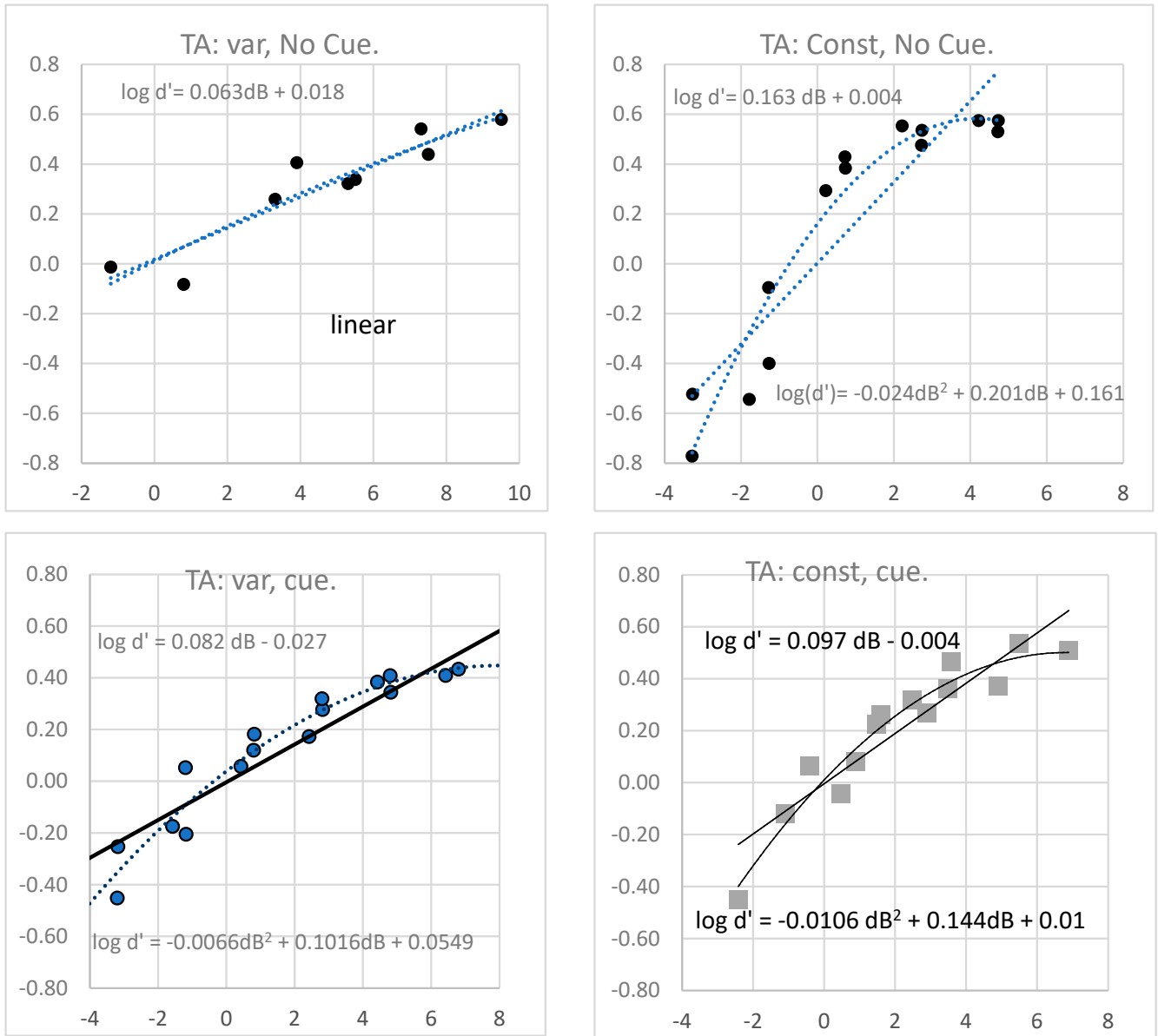

**Figure 2.** $\log_{10}$(d') for all frequencies, plotted against dB SL. Listener TA. Frequency was uncertain in var (**left** panels) or certain in const (**right** panels). There was no cue (**upper** panels) or a cue (**lower** panels). Solid lines show linear regressions; mild bows were fit to the quadratic regressions shown by dotted lines.

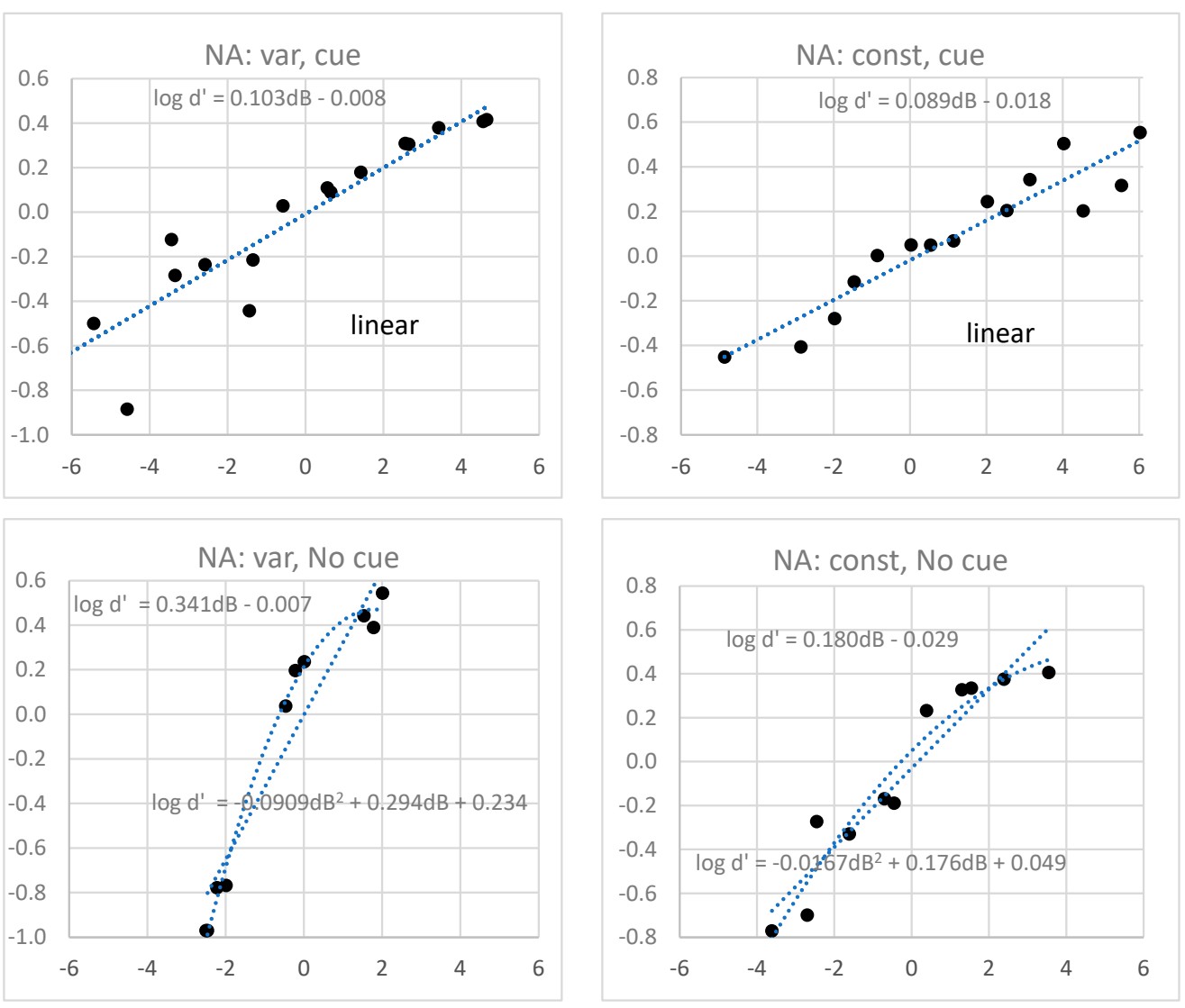

**Figure 3.** As in Figure 2, for listener NA.

**Table 4.** First three data columns: slopes, intercepts, and $r^2$ for linear regressions of log(d′) on signal dB SL for each listener and condition, averaged over the three frequency groups. Last two columns: linear and quadratic $r^2$ for the shifted data shown in Figures 2–4.

| Listener | Condition | Means over Frequencies | | | Shift | Shift |
|---|---|---|---|---|---|---|
| | | slope | inter cpt | $r^2$ lin | $r^2$ lin | $r^2$ quad |
| TA | var No Cue | 0.073 | −2.46 | 0.88 | 0.86 | 0.87 |
| | const NoC | 0.165 | −5.77 | 0.82 | 0.83 | 0.92 |
| | var Cue | 0.084 | −3.08 | 0.89 | 0.86 | 0.98 |
| | const Cue | 0.110 | −4.13 | 0.94 | 0.85 | 0.94 |
| NA | var No Cue | 0.340 | −12.76 | 0.90 | 0.91 | 0.99 |
| | const NoC | 0.195 | −7.05 | 0.95 | 0.89 | 0.93 |
| | var Cue | 0.102 | −3.96 | 0.87 | 0.85 | 0.85 |
| | const Cue | 0.089 | −3.43 | 0.94 | 0.89 | 0.90 |

**Table 4.** *Cont.*

| Listener | Condition | Means over Frequencies | | | Shift | Shift |
|---|---|---|---|---|---|---|
| | | slope | inter cpt | $r^2$ lin | $r^2$ lin | $r^2$ quad |
| MA | var No Cue | 0.099 | −3.57 | 0.85 | 0.69 | 0.81 |
| | const NoC | 0.125 | −4.35 | 0.88 | 0.71 | 0.92 |
| | var Cue | 0.134 | −4.76 | 0.94 | 0.87 | 0.89 |
| | const Cue | 0.109 | −3.94 | 0.91 | 0.83 | 0.94 |
| Mean | var No Cue | 0.171 | −6.26 | 0.88 | 0.82 | 0.89 |

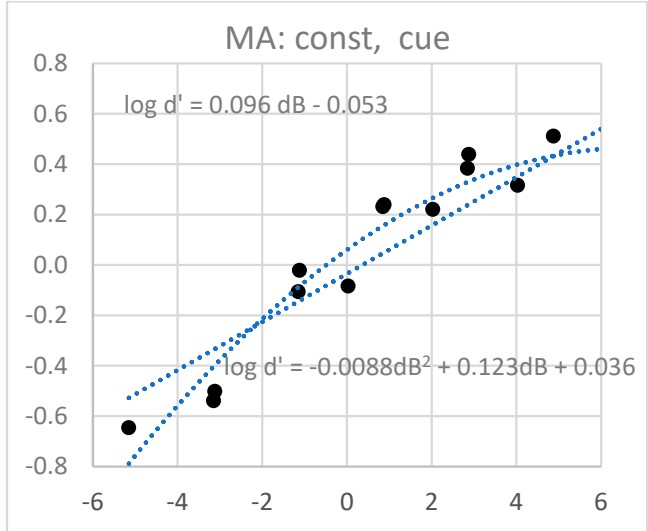

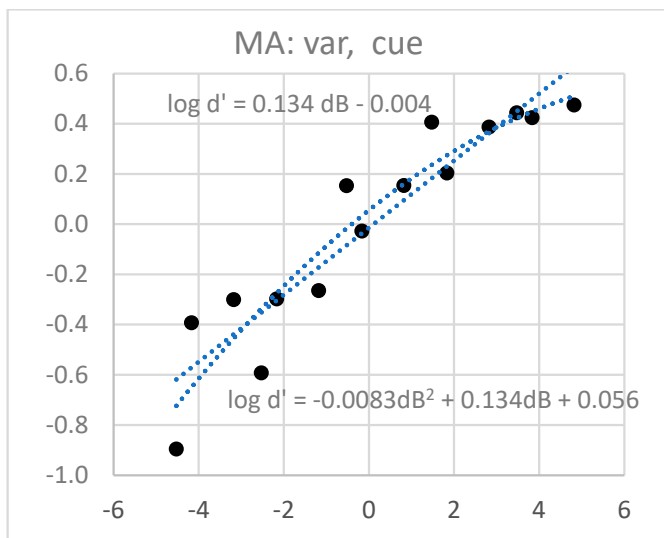

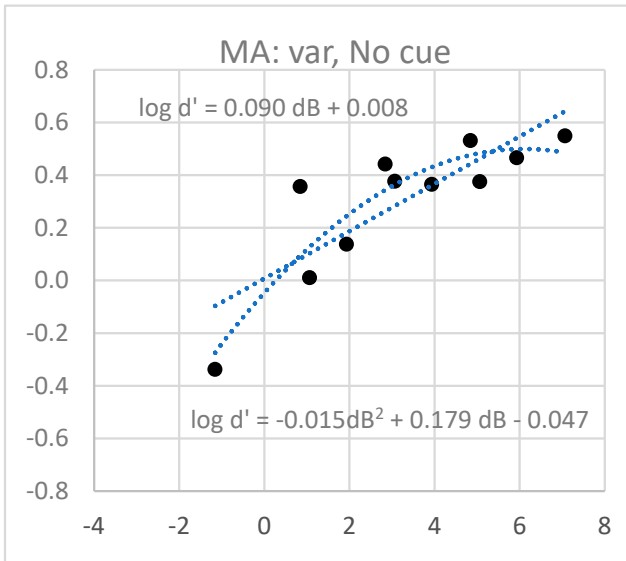

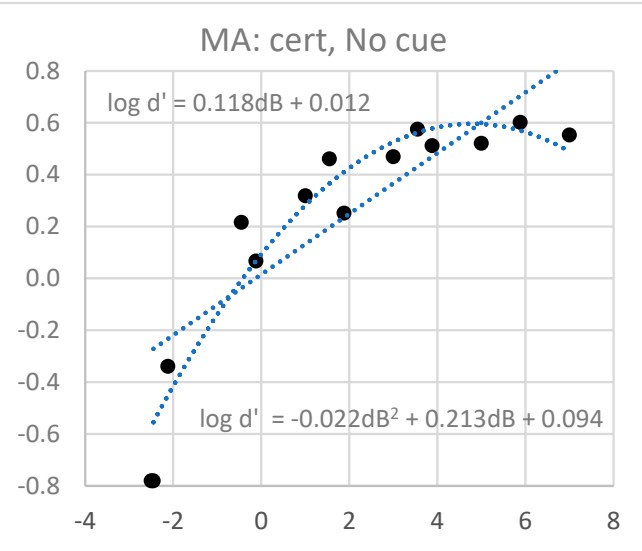

**Figure 4.** As in Figure 2, for listener MA.

A modification of Equation (1) to include a hard threshold, So, helps linearize these bows. Signals below the hard threshold are assumed to be inaudible. The *effective* signal is now defined as the signal level above So, so d′ = 0 if S < So and

$$d' = k(S − So)/No, \text{ for } S > So. \tag{3}$$

The effective signal in dB, namely $20\log_{10}(S − So)$, is shifted further to the left at low than at high dBs, straightening out the bows. Figure 5 shows the quadratic fit to Green's [1]

*const* data (with lapsing accounted for) on the left, and the linearized curve applied to the same data with So = 6 on the right.

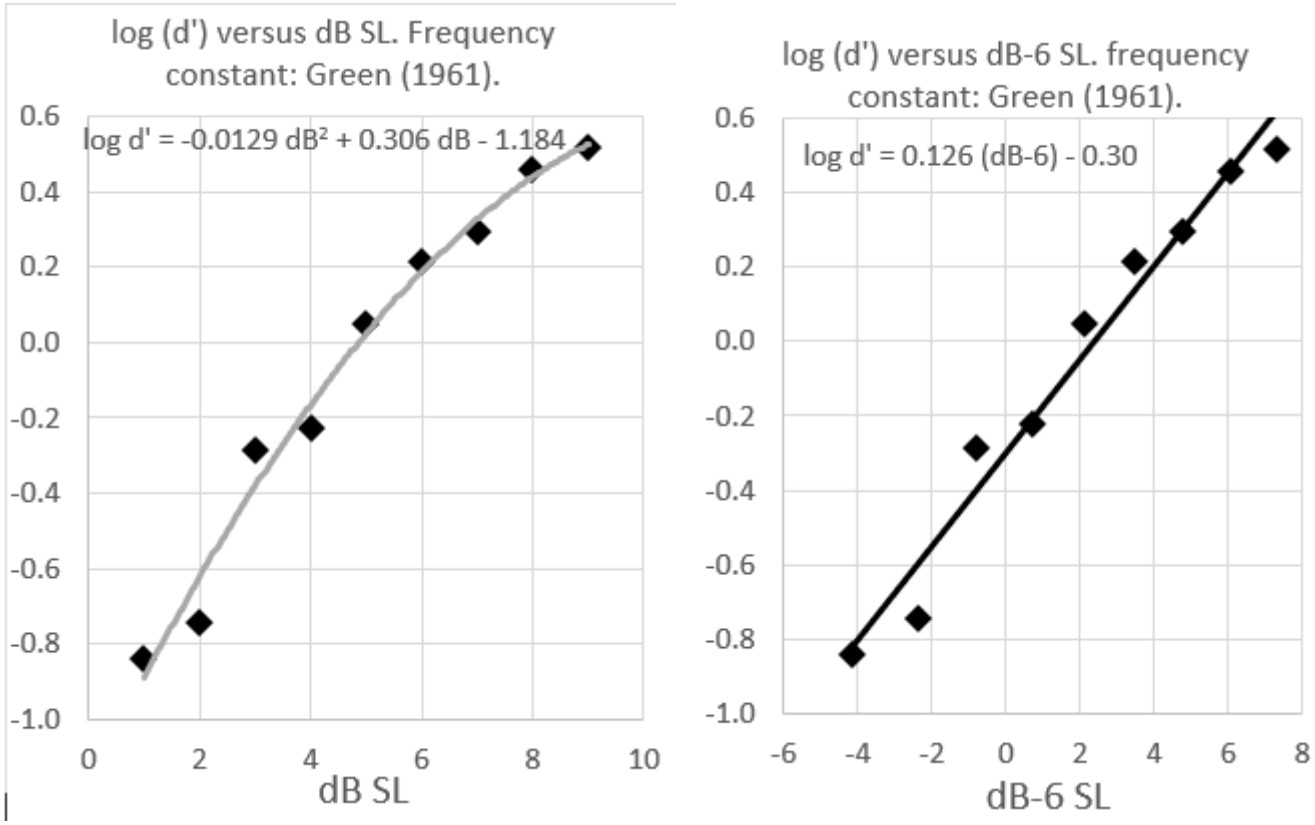

**Figure 5.** *Const* data from Green [1], taken from Table 1 above, plotted as log (d') versus signal dB SL (**left**), and after applying a hard threshold of 6 dB (**right**).

The same approach was taken for the present data (Figures 2–4). A value of So = 2.5, or 8 dB SPL, straightened out the quadratic bows and provided the best-fit linear regressions when applied to all three listeners and conditions. Note that So would be smaller, approaching 0 dB SPL, for the detection of long duration tones at absolute threshold. As Green and Swets [32] point out, ideally signal detection theory presumes that there is no sensory threshold. However, there is no obvious reason for a quadratic function, and a hard threshold may be more realistic.

## 4. Discussion

Listeners differ in the extent to which they benefit from cuing the frequency of the signal and knowing the frequency in advance. One explanation for the individual differences, as presented by Green [1], is that listeners vary in the degree to which they can voluntarily focus on a known frequency. However, there are no independent measurements here of focusing efficiency, so this remains speculative. A follow-up study with more listeners would be helpful in this respect, if the ability to focus could be independently assessed, perhaps using the probe-signal method of Greenberg and Larkin [12] or the overshoot method of Scharf et al. [34] on the same listeners. Here, one can only conclude that for this small sample, pre-cuing 82 ms beforehand *with a valid cue* let these listeners detect signals based on signal energy.

Green [1] compared known frequency to unknown as one way of assessing the effect of knowing the signal exactly by contrasting *const* with *var*. This procedure, also adopted here, may confound exogenous attention, controlled by the stimulus, with endogenous attention, controlled by the expectation or knowledge of the listener. In *const*, the same tone is repeated trial after trial, which can lead to two opposite effects; the priming of one tone

on the subsequent one of the same frequency, which can increase stimulus salience, and inhibition or fatigue due to repeated stimulation, which can reduce the salience of weak stimuli [24]. These exogenous effects do not occur in *var*. Thus, the comparison of *const* with *var* may not be a simple comparison of known frequency versus unknown. Separating these two sources of attentional control would be useful.

The assumption that attention is paid equally to the signal as to the external noise in the signal critical band (the CB) in const is plausible for long-duration tones when the attention band has the same width as the CB. For the brief tones used here, the situation is more complex. Reeves [15] showed that the attention band for 40 ms tones is wider than the signal CB due to additional noise from adjacent CBs, which is progressively removed as signal duration increases. Unexcluded noise will have reduced the uncertainty effect. Scharf et al. [16] showed that, for 350 ms tones, the width of the attention band is equal to the CB whether the signal is cued or not, but no such evidence exists for 40 ms tones. A further issue is that the attention effect inferred here is untethered to other experimental methods of controlling attention. A final problem is that the temporal evolution of the cue and signal was ignored here, yet small differences in timing between the cue and the signal have large effects on signal detection [26]. Indeed, listeners can pick up temporal coherence in complex auditory streams even when paying attention elsewhere [36,37]. Knowing the temporal dynamics of focusing may aid understanding how attention suppresses noise and possibly enhances the signal.

### 5. Conclusions

In conclusion, uncued signals do not follow energy detection but generate steeper slopes, which may be partially accounted for by uncertainty even when stimulus frequency is known. Reducing uncertainty by pre-cuing, so that listeners can focus on the signal frequency and avoid including noise from irrelevant critical bands, demonstrates that the energy model assumed by Green [1] is correct after all. This finding comports well with the fundamental role of attention summarized by Grossberg [7], in which resonance with known information in long-term memory (here, pitch), followed by a successful match, aids perception.

**Funding:** AFOSR grant FA9550-04-1-0244 to Reeves and Scharf.

**Data Availability Statement:** Data are tabulated in the paper.

**Acknowledgments:** Zhenlan Jin programmed the experiments and Jennifer Olyjarchek helped run subjects and tabulate data. The late Bertram Scharf conceptualized the research program but did not plan these experiments or write the paper; any errors or misinterpretations are the sole responsibility of A.R.

**Conflicts of Interest:** The author and lab members have no conflict of interest.

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
