# Peer review of "The Psychometric Function for Focusing Attention on Pitch"

_information, doi:10.3390/info14050279_

Round 1
Reviewer 1 Report
1-- It would help the readability of the manuscript, if a paragraph at the end of the introduction specifies the major contributions.
2-- It would help if one or two numerical results are added to the abstract.
3-- Figures need some cleaning and re-drawing (For instance Fig 5).
Author Response
1-- It would help the readability of the manuscript, if a paragraph at the end of the introduction specifies the major contributions. DONE
2-- It would help if one or two numerical results are added to the abstract. DONE; slope and hard sensory threshold specified.
3-- Figures need some cleaning and re-drawing (For instance Fig 5). DONE. BUT THE PDF IS POOR
Reviewer 2 Report
The written of this paper is poor for reading. the motivation, and idea of this work are unclear.
Author Response
I hope the reviewer finds the revision improved, but there were no substantive comments to address, and the issues were clear to the other reviewers.
Reviewer 3 Report
I think this is a very interesting work, and the field readers are very interested. I agree to publish it.
There is only one minor comment:
(1) The readability of the manuscript is not high. I think it is caused by typesetting and suggest that tables and other forms be unified.
Author Response
Rev. 3 I think this is a very interesting work, and the field readers are very interested. I agree to publish it.
Thankyou !
There is only one minor comment: The readability of the manuscript is not high. I think it is caused by typesetting and suggest that tables and other forms be unified.
The pdf seems impossible to fix. Fig. 5 is improved, but the layout is still poor.
Reviewer 4 Report
This article analyzes and discusses relevant psychophysical data on the processing of auditory information, here pitch (energy), by the human brain. Pitch in the sensory physics domain refers to the position within the frequency (energy) domain of a single sound in the complete range of sounds of a wave. Sounds will have higher or lower physical energy (pitch) according to the frequency of vibration of the sound wave producing them. It is shown here that focussed attention improves the perception of auditory pitch in human observers. The experiments are well-conceived, the psychophysical data shown are sound. The findings are placed within the context of state-of-the-art empirical and theoretical psychophysics. They support an important claim of Grossberg's Adaptive Resonance Theory (ART) by showing that early and not necessarily conscious mechanisms of attention can influence the accuracy of our perceptions. The article makes a valuable contribution to the Special Issue.
The author should perform the following minor revisions:
1. Pease adapt the references to the format of the journal
2. Please break down the sections Discussion and Conclusions into two; do not start the Discussion section with a subtitle, which should be preceded by a short statement regarding what will be discussed before giving the first subtitle. Also, if this first one is to be the only subtitle of your discussion, remove it altogether.
Congratulations to a very nice paper!
Author Response
This article analyzes and discusses relevant psychophysical data on the processing of auditory information, here pitch (energy), by the human brain. Pitch in the sensory physics domain refers to the position within the frequency (energy) domain of a single sound in the complete range of sounds of a wave. Sounds will have higher or lower physical energy (pitch) according to the frequency of vibration of the sound wave producing them. It is shown here that focussed attention improves the perception of auditory pitch in human observers. The experiments are well-conceived, the psychophysical data shown are sound. The findings are placed within the context of state-of-the-art empirical and theoretical psychophysics. They support an important claim of Grossberg's Adaptive Resonance Theory (ART) by showing that early and not necessarily conscious mechanisms of attention can influence the accuracy of our perceptions. The article makes a valuable contribution to the Special Issue.
Thankyou for this very positive and informed comment !
The author should perform the following minor revisions:
- Please adapt the references to the format of the journal
I have revised the pdf and numbered the references .
- Please break down the sections Discussion and Conclusions into two; do not start the Discussion section with a subtitle, which should be preceded by a short statement regarding what will be discussed before giving the first subtitle.
DONE
Also, if this first one is to be the only subtitle of your discussion, remove it altogether.
DONE
Congratulations to a very nice paper! THANK YOU
Round 2
Reviewer 2 Report
The paper written is very poor for reading. Besides, the technical novelty and depth of this work are very limited. Thus the quality of this paper is far away for accepted.